# "I Keep Sweet Cats In Real Life, But What I Need In The Virtual World Is A Neurotic Dragon": Virtual Pet Designs With Personality Patterns

Hongni Ye*
Duke Kunshan University

Ruoxin You†
University College London

Kaiyuan Lou‡
Duke Kunshan University

Yili Wen§
Duke Kunshan University

Xin Yi¶
Tsinghua University

Xin Tong‖
Duke Kunshan University

## ABSTRACT

Virtual Pets serve as companionships and meaningful in-game narratives in the metaverse. Players have unique personalities and personality preferences for their pets. However, the design of virtual pets often relies on designers' individual experiences without considering the virtual pets' personalities. We designed the virtual pets' visual appearances by following the design guidelines from the Five Factor Model (FFM) in voxel format. We conducted a study to investigate people's perceptions of virtual pets' personalities and appearances through two user studies. Our findings suggest that voxel-style virtual pets better represented agreeableness than realistic pet pictures. Additionally, users prefer virtual pets (voxel-style pets generated with machine-learning techniques) that share similar personalities. The study's results provide valuable insights for game designers and researchers for future pet game design and understanding of how people perceive virtual pets based on their appearance and behavior.

**Index Terms:** H.5.2 [User Interfaces]: User-centered design—Style guides; H.5.1 [Multimedia Information Systems]: Artificial, augmented, and virtual realities—Evaluation/methodology

## 1 INTRODUCTION

As technology advances, the demand for artificial companions [14] with diverse personalities in different social roles increases [13]. Virtual pets, in particular, have been gaining immense popularity in the digital games field. They serve as a source of entertainment [11, 48, 67] and present the potential to promote mental health [9, 58, 90] and enhance children's skill development [8, 12, 24, 45]. Virtual pets offer a unique opportunity for individuals who cannot keep real pets for various reasons, such as allergies or lack of resources. Moreover, research has suggested that interacting with virtual pets can lead to positive emotional outcomes, such as reduced anxiety [33].

There are various styles for designing virtual pet characters, the most popular ones are the realistic style and cartoon style. The realistic style, in high fidelity, provides a lifelike appearance [58], while the cartoon style, including voxel and two-dimensional sketch, creates a more cartoon look. For example, Tamagotchi [4] and Pokemon Go [2] use sketch and cartoon 3D styles, respectively, and 3D voxel-style virtual pets are present in games like Minecraft [1] and Sandbox [3]. To ensure efficient character modeling and avoid negative impacts on people's perceptions of virtual characters due to low aesthetic qualities and intermediate rendering realisms [73], we

*e-mail: hongni.ye@mail.polimi.it
†e-mail: echoyou67@outlook.com
‡e-mail: midstream.lou@gmail.com
§e-mail: yili.wen@dukekunshan.edu.cn
¶e-mail: yixin@tsinghua.edu.cn
‖e-mail:xt43@duke.edu

designed our virtual pet characters in voxel style. With the explosive growth of machine learning and the refinement of generative networks, neural networks like Generative adversarial networks (GAN) or diffusion models are widely used to replace the manual production of pictures and models in the industry [25, 65]. However, generative models are more common in 2D pictures or 3D facial construction than virtual pets, making pet models scarce and expensive.

Virtual pet games lack personality diversity and fail to provide a similar experience to raising a real pet, as their personalities are unrelated to their appearance. Previous studies have explored personality differences in real pets, such as cat and dog breeds [15, 50, 71, 87]. Although previous studies have shown that personality differences exist in non-human animals [27], it lacks research on cross-species personality traits in pets. This gap in the literature limits our understanding of the factors that contribute to the development of personalities in different species and how we can design virtual characters that can mimic and respond to these traits.

To address the gaps in current research, our study focuses on creating virtual pets that display variations in personality and exploring how players perceive their behaviors and appearance. We aim to answer three primary research questions: (1) How do users perceive the personalities of virtual pets in different styles and representations? (2) How do virtual pets' personalities relate to their appearance, and how can we design virtual pets involving their personalities? (3) What are individuals' perceptions of the appearance and personality of virtual pets generated through machine learning techniques?

Through two studies, we found that style and presentation significantly affected users' perceptions of virtual pet personalities. Additionally, we found that visual cues, such as skin color and body shape, can significantly influence how virtual pets' personalities are perceived. In general, our contributions will be threefold: (1) In the game design domain, we applied pets' personality variations to virtual pet design by following the FFM. And we evaluated our designed virtual pet characters with pre-defined personality traits with users' study. (2) We combined the traditional method and the machine learning technique to generate the appearance of virtual pet. We embedded Neural Cellular Automaton (NCA) [55] into our generation process to increase the diversity of generated models, which provides a novel method for game character design. (3) Our study examined the potential of using voxel pets' appearances as virtual companions to enhance the mental well-being of young individuals by reducing anxiety levels through interactive engagement with virtual pets.

## 2 RELATED WORK

Below, we describe the design examples, and machine learning techniques for generating models of virtual pets.

### 2.1 Virtual Pet Design and Generation

Pet serves the important function as a human companion, a strong and healthy human-animal relationship will be beneficial for both entities [82]. As suggested by previous researchers, appearance is an important consideration in pet owners' decision-making [86]. And

One tenant of folk psychology is that people tend to select pet dogs that have a similar appearance to themselves [68]. And in the artificial animal design field, a study about human–alloanimal relations highlighted that cartoon animals can lead to people wanting to be close to the depicted animal, for the reason that their appearances are designed as approachable, cuddly, friendly, and fun beings [18]. Thus, the appearance design deserves our attention when conducting virtual pet design work.

In the Virtual Reality (VR) pet game domain, to promote better game experience and immersion, pets' appearance design has an intimate connection with user preferences. As introduced by Chaoran Lin et al., there are three user types based on user's motivations and expectations: (1) pet-keepers, (2) animal teammates, and (3) cool hunters [48]. The user type model inspired us to do case studies of virtual pets' appearance according to their target user types. And we proposed three virtual pets types: (1) natural; (2) intelligent; (3) fantasy. And we conduct case studies based on the different types, and the results are summarized in cards. See the case study results in the appendix. Through the case study, we discovered most cases are about natural pets and took the accessibility of pets' personality knowledge into consideration, and we decided to focus on the natural pets' design as our target pet type in this study.

Machine learning techniques have gained widespread image and model generation adoption in recent years. Neural networks, including VAEs [42], DCGANs [66], and the diffusion model [36, 77], have demonstrated impressive image generation capabilities. These models have also been extended to 3D model generation by increasing the dimension of the data [65, 91]. Generating 3d mesh is another popular approach to 3D generation, which can also be accomplished with neural networks, such as VAEs [23]. Mesh-based models offer an alternative means of 3D representation that can be more efficient and suitable for certain applications. However, all of these generation approaches require a large amount of high-quality training data, which restricts their usage in Virtual Pet generation. So, we finally decided to use the Neural Cellular Automaton (NCA) in generation [55]. By combining the NCA with the traditional generation process which recombines the different parts of models, we can get plenty of high-quality models with few training data.

## 2.2 Exploring Personality Traits in Animals

As illustrated by Yerkes [93], it is commonplace to regard individual animals as possessing distinct personalities. And other researchers have proved that personality differences do exist and can be measured in animals other than humans [27]. There are a great number of works that studied pet's personalities, including cats [44], dogs [20], ferrets [78], dolphins [57], and other reptile animals [85]. Through the literature review, we discovered the breeds of cats and dogs had been investigated the most. Thus, we further researched the work about personality traits among cat and dog breeds. For the cat breeds, Milla Salonen et al. suggested that cat breeds grouped into four clusters by analyzing their personality traits with three components named aggression, extraversion, and shyness [71]. Other researchers ranked dog breeds on ten behavioral characteristics in three factors (aggression, reactivity, trainability), considering breeds of the three most closely related groupings, the wolf-like, guarding, and herding groups [29]. And refer to Fédération Cynologique Internationale (FCI) [37], there are ten dog groups based on various discriminators such as appearance or role. The existing variations of personality traits among different cat and dog breeds motivated us to apply personality analysis to virtual pet design.

The Five-Factor Model (FFM) is one of the most commonly used instruments for measuring personality for humans [53]. The FFM comprised the dimensions Neuroticism (N), Extraversion (E), Openness to Experience (O), Conscientiousness (C), and Agreeableness (A). And some researchers also applied the FFM to different species' personality tests. For example, Samuel D et al. built a preliminary

framework with the human Five-Factor Model plus Dominance and Activity for measuring the personalities of 12 nonhuman species, and their results indicated that various primates, nonprimate mammals, and even guppies and octopuses all show individual differences that can be organized along dimensions akin to E, N, and A. [26]. And another work presented the " Feline Five, "which was adapted from FFM with a 5-factor analysis: Neuroticism, Extraversion, Dominance, Impulsiveness and Agreeableness. And the "Feline Five" has been proven to introduce a more comprehensive overall pet domestic cat personality structure. Therefore, the "Feline Five" has great potential to measure our designed virtual pets' personalities. We adapted these instruments in our pet design method, which will be introduced in section.4.

## 3 VIRTUAL PET DESIGN IN VOXEL STYLE

To investigate the influence of pets' appearance traits on human perception of personality, we developed virtual pet characters inspired by real-life pets. Our methodology involved categorizing primary cat and dog breeds into six distinct clusters based on their appearance traits and creating a virtual pet character to represent each cluster. Using this approach, we aimed to make virtual pets with unique and recognizable physical characteristics while incorporating diverse appearance traits.

### 3.1 Design Objective

To address the research questions, our design objective was to create virtual pet characters with a broader range of appearance traits and associated perceived personality traits. We intended to develop a mapping guideline that linked the appearance traits of the pets with their corresponding personality traits. As such, we hypothesized an appearance-personality mapping to guide our design process. We recognized that individual perceptions might vary. Hence we aimed to ensure a degree of consistency in the perceptions of people towards our virtual pets.

### 3.2 Design Baseline: Clusters of Cats and Dogs

After conducting a case study on three types of virtual pets (see Fig.**??**, we decided to begin our initial design with dogs and cats, among the most popular and common domestic pets [5]. To achieve our design objective of creating virtual pets with a broader range of appearance traits and perceived personality traits, we compiled a list of common domestic cats and dog breeds. We categorized them into clusters based on the distinctive characteristics of different body parts. In doing so, we referred to the classification systems of FCI [37] and Cat Breeds [62]. We then mapped personality traits to different breeds to identify consistencies within each cluster. Personality traits were transferred from previous studies on cats' and dogs' behavior traits [30, 72]. We merged groups with similar appearance and behavior traits while excluding those with various behavior traits, dividing the different breeds of cats and dogs into six clusters. Fig. 1 displays the appearance traits, typical breeds, hypothesized personalities, and our corresponding virtual pet characters for each cluster.

### 3.3 Design Baseline: Visual Expression

In addition to character archetypes, visual expression plays a crucial role in shaping people's perception of virtual pets [81]. To ensure that our designed pets are displayed in a multi-dimensional form and have a wide range of appearance traits, we created them in 3D using voxel style. This approach increases our efficiency in character modeling and eliminates the potential negative impact of low aesthetic qualities and intermedia rendering realism on people's perceptions of virtual animal characters [73]. In addition to these benefits, we were inspired by the potential of 3D voxel-style modeling in virtual pet design. We aimed to extend 3D generation techniques into this field, building on the success of voxel-based models in Minecraft

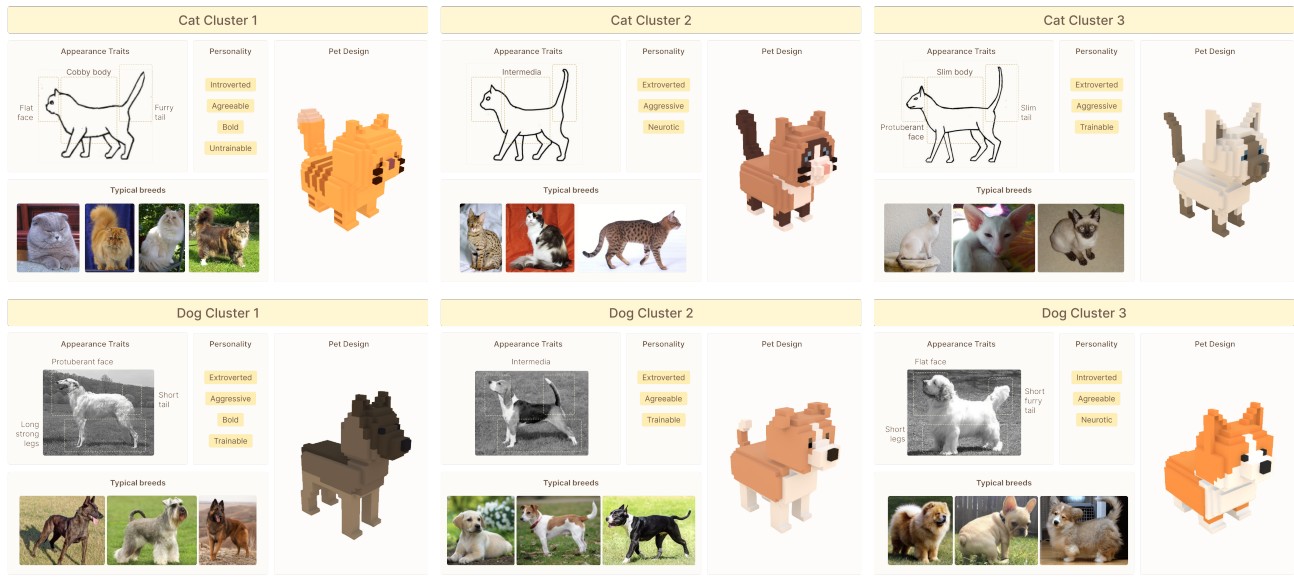

Figure 1: Design of Cat and Dog Clusters. The pet side view pictures used for reference were obtained from [31] and [32].

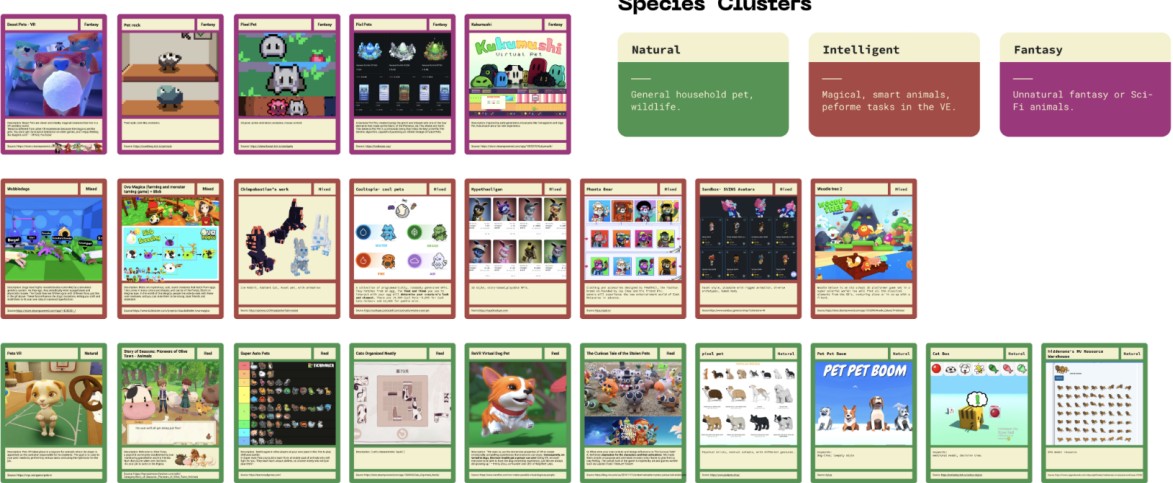

Figure 2: Case study clusters, (1) natural pets (in green cards): the virtual pets who have a similar appearance and characteristics as the real-life pets, such as dogs, and cats; (2) intelligent pets(in red cards): good teammates for players, who can perform tasks in the virtual environment; (3) fantasy pets(in purple cards): have sci-fi look, and can assist players in exploring in the environment.

and sandbox games. 3D generation techniques would be more capable of generating visually appealing virtual pet characters. During our follow-up interviews, we found that people preferred virtual pets in voxel style compared to realistic models, further confirming the potential of the voxel style.

Visual elements, including shapes, volumes, and colors, are essential components of character design that significantly impact creating emotional experiences [76]. To avoid potential influence on the diversity of perceived personality traits between realistic and virtual pets, we controlled the visual elements by following realistic dogs and cats' body structure, proportion, and color palette in our character designs. This approach ensures that any differences in perceived personality traits between realistic and virtual pets can be attributed to factors other than visual expression.

## 3.4 Character Design

We created six characters based on the clusters, selecting one breed within each cluster as the model sample. We used Magicavoxel[1] to create voxel-style characters and render static pictures. We limited the model sizes to 31x31x31 voxels to facilitate pet generation.

In addition, previous research showed the potential impact of animation on people's perception of virtual pets [75]. Therefore, we designed another version of the virtual pets with natural movement to study the effect of additional expressiveness on people's perceptions. We did not involve facial animation because of the potential negative reaction caused by the animal uncanny valley [73]. We designed a walking animation for the cat clusters and a running animation for

---

[1] https://ephtracy.github.io/

the dog clusters based on the nature of these two pet species. We used VoxEdit[2] to build and Blender[3] to render the animation clips.

## 4 STUDY 1

Study 1 included surveys and interviews to understand users' perspectives of virtual pet characters. The survey was designed with three goals: 1) compare participants' perceived personalities of different styles of pets within the same cluster. 2) evaluate how our designed pet attracts people. 3) understand their perception toward keeping real pets and virtual pets. The follow-up interview aimed to understand further the reasons behind participants' perceptions based on the result of the online survey. The study obtained ethical approval for the study from the Institutional Review Board.

We designed the survey using a mixed 3x6 design (pet character styles * pet clusters). Our experimental conditions of the pet character styles included: real static pets, static virtual pets, and animated virtual pets. We controlled the pets representing different conditions belonging to the same clusters, which we defined in section 3.3. Participants were randomly assigned one pet cluster and rated personalities and overall feelings of all three pet character styles. We then invited 9 participants for follow-up interviews to determine the factors of people's perceptions. The interview questions were about the factors of participants' answers, and two card-sorting sessions for further exploring participants' perceptions based on their answers.

### 4.1 Participants

Participants voluntarily self-selected to complete the survey and consented before taking it. We recruited 33 participants (12 males, 20 females, 1 non-binary) via social media and word of mouth. Participants were 18-24 (N=24) and 25-34 (N=9). 24 participants had the experience of keeping real pets, 7 had dogs, and 7 had cats. For virtual pets, 13 participants reported they had played virtual pet games before, including Animal Crossing (N=3), Tamagotchi (N=2), Tencent QQ Pet (N=6), and others (N=4).

### 4.2 Measurement

The online survey consists of four parts. The first three parts measured the perceived pet personality under three conditions. One real pet picture, one static virtual pet picture, and one virtual pet animation clip were randomly distributed into one of the three parts. All real pet pictures were downloaded online, with the same white background and showing the pet's whole body. We downloaded ten pictures for each cluster and randomly displayed one on the survey. The researchers created static pictures and animation clips of virtual pets.

We measured the perceived personality with an adapted 7-point scale. We designed the scale based on the Feline Five [49], and pooled items from previous personality assessments on cats [47] and dogs [70]. The scale included four measurement dimensions which were general and commonly used to measure pets' personalities. Each dimension had four pairs of contracting description items. All 32 items showed in random order in each part of the survey. Participants rated each item from 1 to 7 points according to what extent they agreed with the description corresponding to the material provided. After the 32-item chart, two questions followed to figure out whether the participants knew the pet's breed in the picture and their overall feeling about this pet. The last part accessed participants' demographic information, experience keeping pets, and attitudes toward pets.

The semi-structured interview has four themes: pet personality, pet appearance, pet interaction, and feelings of our designed pet characters. We collected participants' survey answers and visualized

---

[2]https://www.voxedit.io/
[3]https://www.blender.org/

them in a table shown during the interview to recall their memory. Besides, we created two card sorting sessions based on the data of open-end questions. One aimed to determine participants' perceived cuteness, which had primarily been proposed as an expected feature of virtual pets from the surveys. We created the cards with our designed and similar voxel characters and asked the participants to select the cards they thought were cute. The other card sorting focused on the expected interaction methods towards virtual pets, which had been asked on one of the open-end questions in the survey. We coded participants' answers and selected nine of them to make cards. We asked the participants to pick and rank the cards according to their expected interactions.

### 4.3 Procedure

Participants were randomly distributed into one of six control conditions and completed a Qualtrics form. On the last question, we asked if they were willing to participate in our follow-up interview. After analyzing the data of the surveys, we emailed ten participants whose answers were consistent or contrary to our data results. Nine of them consented to take the interview. Our interviews were conducted online via the Feishu meeting. The participants first had five minutes to read and sign the consent form. Then, we conducted a 40-min semi-structured interview with our participants; participants received 100 RMB as compensation for their time and contribution.

### 4.4 Results

The quantitative and qualitative results showed that the style and appearance of virtual pets significantly impact participants' perceptions of their personalities. We also found that participants connected perceived personalities with the appearances of virtual pets. We concluded that the design suggestions involve expected pet types, personality presentation factors, and interaction with pets, which benefit future virtual pet design.

#### 4.4.1 Perceived Personality in Different Styles

The results of the repeated-measures ANOVA test shows that the style of pets (realistic, virtual) and the presentation (static, animation) significantly affected people's perceptions of their personality traits, especially for Neuroticism and Agreeableness. Fig 4 shows the distribution of scores on four personality traits. People's perception of agreeableness is primarily influenced by pet styles ($F(2,29) = 8.10, p < 0.01$). The results indicated that participants thought voxel pets are much more agreeable ($mean = 40.58, SD = 6.95$) than realistic pets ($mean = 33.7, SD = 8.42$). For voxel animations, they received an agreeableness score close to static voxel pets($mean = 37.7, SD = 7.82$). For Neuroticism, realistic pets receive the highest score ($meaN = 31.6, SD = 5.74$), and both static voxel pets and voxel animations are less neurotic (static voxel pets: $mean = 26.8, SD = 5.42$, animation: $mean = 27.87, SD = 5.77$,). Participants' perceptions of pets' extraversion are not significantly influenced by pets' style ($F(2,29) = 1.28, p = 0.28$). All three style receives extraversion score relatively close to each other (realistic pets: $mean = 36.67, SD = 7.61$, static voxel pets: $mean = 33.73, SD = 7.47$, animation: $mean = 34.5, SD = 5.45$). Impulsiveness keeps stable when the style changes($F(2,29) = 0.59, p = 0.55$). The low standard division illustrates that most participants give similar scores to all three styles (realistic pets: $mean = 36.67, SD = 7.61$, static voxel pets: $mean = 33.73, SD = 7.47$, animation: $mean = 34.5, SD = 5.45$).

Through the interview, we discovered three main reasons for these results: Firstly, the design details played a significant role in expressing agreeableness (N=8). For example, a participant evaluated the voxel pet with high agreeable scores explained to us, *"I might think voxel is a little bit chubby and silly, but I think it's a little more friendly"* (P15). Secondly, their decision-making for the evaluation

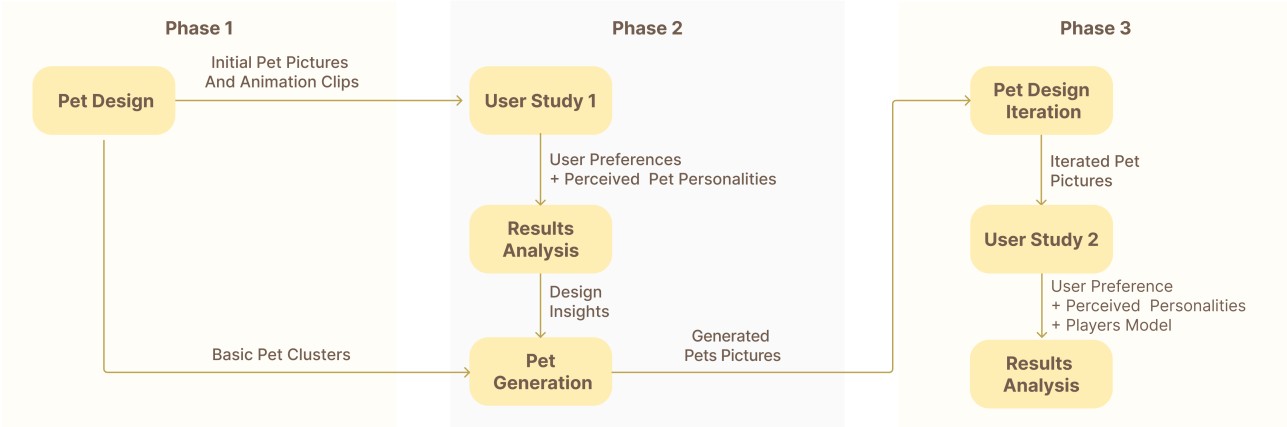

Figure 3: Procedure of Study1 and Study 2

was influenced by their previous experience and was related to individual differences in spending time with pets (N=3). A participant rated the realistic pets with the highest score in neuroticism and said *"If it was a real pet, it reminded me that it could do things that were threatening to me. I was bitten by a dog when I was a child, whereas these virtual things couldn't really threaten me."* (P25). Thirdly, some participants rated their personality by inferring the pet breeds (N=3). P33 told us, *"The real pet photo is of a Muppet cat, albeit well-behaved. But voxel's static and animated features are not reminiscent of a cat running amok."* Thus, we could understand why voxel has the best effect for showing agreeableness, while real pet pictures make the pets look more neurotic.

### 4.4.2 Association between Personality and Appearance.

Our goal was to create pets with various personality types based on appearance and perceived personality traits, resulting in six pet clusters, as depicted in Fig. 1. Our survey results indicated that participants' perceptions of pets' personalities aligned with our design intentions. For instance, participants rated the first cat cluster (fat yellow cat) and the second dog cluster (medium-sized yellow-white dog) as having the highest agreeableness scores, consistent with our pet personality design goals.

Moreover, our analysis of participants' extroversion rankings for the cat and dog clusters aligned with our initial design intentions. Specifically, participants' extroversion rankings for the cat clusters were in the order of cat cluster 2, cat cluster 3, and cat cluster 1. In contrast, their rankings for the dog clusters were in the order of dog cluster 3, dog cluster 2, and dog cluster 1. These results suggest that our virtual pet design successfully conveyed various personality traits through appearance and that participants could perceive and rate these traits accurately.

### 4.4.3 People prefer pets with cute appearances but neurotic personalities.

Cuteness was the word repeatedly mentioned by participants during the interview. We received the answers for why people like cute pets because cute looks make people feel safe, and another explanation is that pets that look cute are easier to keep. For instance, one participant liked cut pet and explained to us, *"Cute looking pets are better behaved and easy to keep, while naughty pets may be more nerve-racking."* (P16). Cuteness has some common distinct appearance patterns, one is the body shape. Participants associated these traits with cuteness: small size, short and fat, short legs, and round ear shape. *"The pet's small size makes it less aggressive"*, the quote by P4. And *"The ears of this pet are round, which makes me think she is very friendly."* said by P16. Another key factor that

makes pets look cute is the coat color. As mentioned by interviewees, warm and bright colors, heterochromous and clean colors are the color of cuteness in their minds. And other factors that contribute to a pet's cuteness consist of special shapes, such as a lighting-shaped tail. And the pets with beards and dimples are definitely plus points for being cute. All interviewees treated cuteness as a sign of agreeableness; however, some participants preferred a contrast between personality and appearance (N=5). Specifically, they like virtual pets who have cute appearances but are inclined to be neurotic and cold in personality. One interviewee told us, *"I like those crazy pets. They're more neurotic. Because animals don't do it like that, you might have difficulty understanding it, and there's a great sense of mystery."* (P16).

In a word, we intended to construct a relationship between virtual pets' appearances and personalities. Through the user study, we proved the pieces of evidence we showed through our design work. And we found all interviewees thought of cuteness as a sign of agreeableness. Further, we also discovered that people prefer virtual pets with cute looks but with neurotic personality traits.

### 4.4.4 Suggestions for Virtual Pet Type and Interaction Design in Pet Game

Through the open questions in the survey, we found 3 participants expected to keep fantasy pets, such as dragons, dinosaurs, and sci-fi pets that can not be found in real life. And 7 participants expected to keep cats and dogs. We asked why they decided on the expected pet type selection, and we discovered some people have experienced petting a pet type, and (s)he (N=1) decided to continue to have the same type as a virtual pet. However, other participants do the opposite. We conclude with two main reasons for this. One is that pet keepers recall their memories with pets, this could be both positive and negative, which leads to their decision-making on keeping the same pet type or not. Another reason is mainly about the specific pet personality traits; that is to say, people would be addicted to certain personality traits of pets, such as agreeableness, as a consequence, they regard pets who possess these traits as the first choice.

We also discovered interesting findings about pet animation through the interview. On the one hand, the animation of the pet combined with the sound can convey its personality more directly. There is a quote from P4: *"I think animation is very important for the expression of personality, especially the voice, when it is happy and when it is angry, and when it is threatening, the voice is completely different."* On the other hand, some movements of specific body parts, for instance, ear, tail, and leg movements are important references to the perception of personality (N=3), one interviewee told us, *"I think sometimes the tail of a dog is more informative, that*

*is, you can tell if he is happy or unhappy by his tail, so you can tell what kind of mood he is in, maybe he is extroverted."*(P16).

In addition to the pet type, we investigate the expected interaction with virtual pet in the virtual world. Through the survey's open questions and interview, we concluded the three most popular interactions: Talking(9 votes), touching(6 votes), and feeding(6 votes). And "why do you want to talk with your virtual pet in the virtual pet as the main interaction" is that they want to communicate in the same language to understand the pet better. For example, *"I want to be able to talk to him and he understands and it can react. I'm more interested in the behavior of his feedback than the content of the conversation."* (P25). Moreover, other popular interaction types among the participants are treasure hunting and adventure.

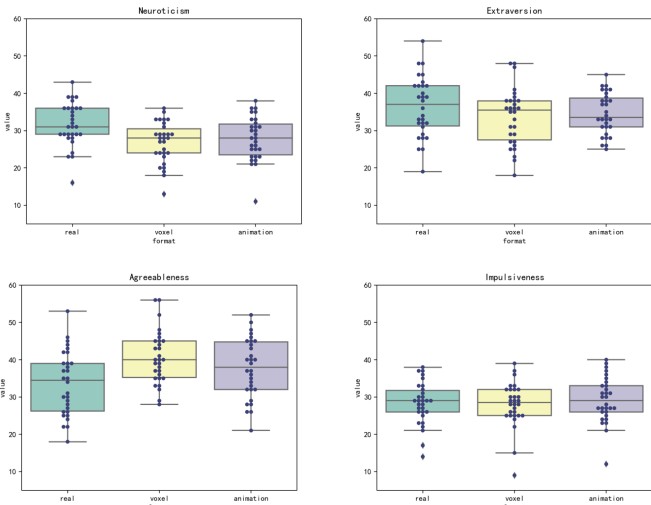

Figure 4: Distribution of scores on four personality traits from 30 people.

## 5  PET GENERATION

We have developed a unique 3D model generator that automatically creates virtual pets by hybridizing existing models and creating new ones based on input. The following section will explain the generator's process, which includes dividing and recombining models, dyeing them for harmonious colors, and texturing them for unique patterns.

### 5.1  Generation

With the development of our search, the demand for a 3D model generator was increasing. Firstly, the number of manually created pet models is too small, and more models are needed to prove the universality of the research outcomes. Then, many models must target which feature, color, or combination leads people's impressions towards a pet more precisely. Moreover, it is also an exploration of generation techniques since the outstanding generation models are mostly 2d-based nowadays, while the demand for 3D models is increasing rapidly. So, we implemented a generator to generate virtual pets automatically. The generator takes input from several 3D pet models and outputs new models based on the input.

The generator uses existing hybrid models and creates new models based on them. Input models are parent models, and generated models are children. Every child model inherits appearances from all their parents (in the generation, a child model can have more than two parent models). Random mutations are applied in the hybrid process to ensure that every newly generated child model is unique

even if they have the same parents, guaranteeing variety and a more significant number of generated models.

The generation process is shown in Fig. 5, which is mainly three steps. Divide and recombination, dyeing, and texturing.

#### 5.1.1  Divide and recombination

The divide and recombination process gives basic shape to newly generated models.

In the research, the generator takes input from three cat models and divides each into five parts: the head, ears, body, tail, and limbs. The generator will loop over all the possible combinations to get the shapes of new models as much as possible. When combined, all the parts will be aligned automatically since the models' size and transformation might differ.

After getting all the possible combinations, the generator will rate the results and pick the reasonable ones to send to the next step. For example, the combination won't give the effect of a cat with the fattest body and thinnest tail, which didn't make sense in real life.

#### 5.1.2  Dyeing

After dividing and recombination, many models with reasonable shapes were generated. But, we can not directly use these models because they might have unharmonic colors. Usually, the inherence of color follows some rules. The child is more likely to have a mixture of color or shows a transition color of its parents. However, the models generated after recombination inherit all the colors and patterns on their parents' skin. A cat can have different colors on their head, body, and limbs, and there is no transition, which makes those models unreal. So, all the models will be repainted after recombination. They can have a color closer to one of their parents or in the middle of their parents' color.

During the dyeing, the generator will give random mutation. After generating the overall color and choosing the palette, the color might mutate several degrees darker or brighter. Also, the larger mutation that gives a model a new palette will happen sometimes. It can prevent models from losing color variety after often dyeing.

#### 5.1.3  Texturing

Texturing is a random process that can give models unique patterns or textures on the skin.

During texturing, each time a new model was generated after the previous two steps, a new model called "mask" will be generated. The masked model is a real-time generated random voxel model of the same size as the generated pet model. Every newly generated pet model will have a mask for it. Then, the pet and mask models will be put in the same coordinate. After that, the generator will iterate every voxel on the model, if there is an overlap between mask and pet, the color of that voxel will change.

The different masks can give different patterns to a pet model. For example, the mask of many little floating balls can make a pet spotty, and the mask of many vertical planes draws stripes on the pet. Since the mask is randomly generated, every pet can have its unique pattern.

Neural Cellular Automaton (NCA) was used to generate masks. The 3D NCA model used in the generator takes the input from the voxel 3D model and also outputs voxel 3D models. The network was trained to generate simple objects like spheres or plates from a single dot as a seed. When generating the mask, a random seed(many randomly distributed dots in 3D space) will be sent to the pre-trained network. Then dots will start to grow to the object when running the network. The growing process will stop after random steps, then the output model will be the final mask.

## 6  STUDY 2

We conducted Study 2 through surveys and semi-structured interviews to gather users' feedback on our generated characters. This

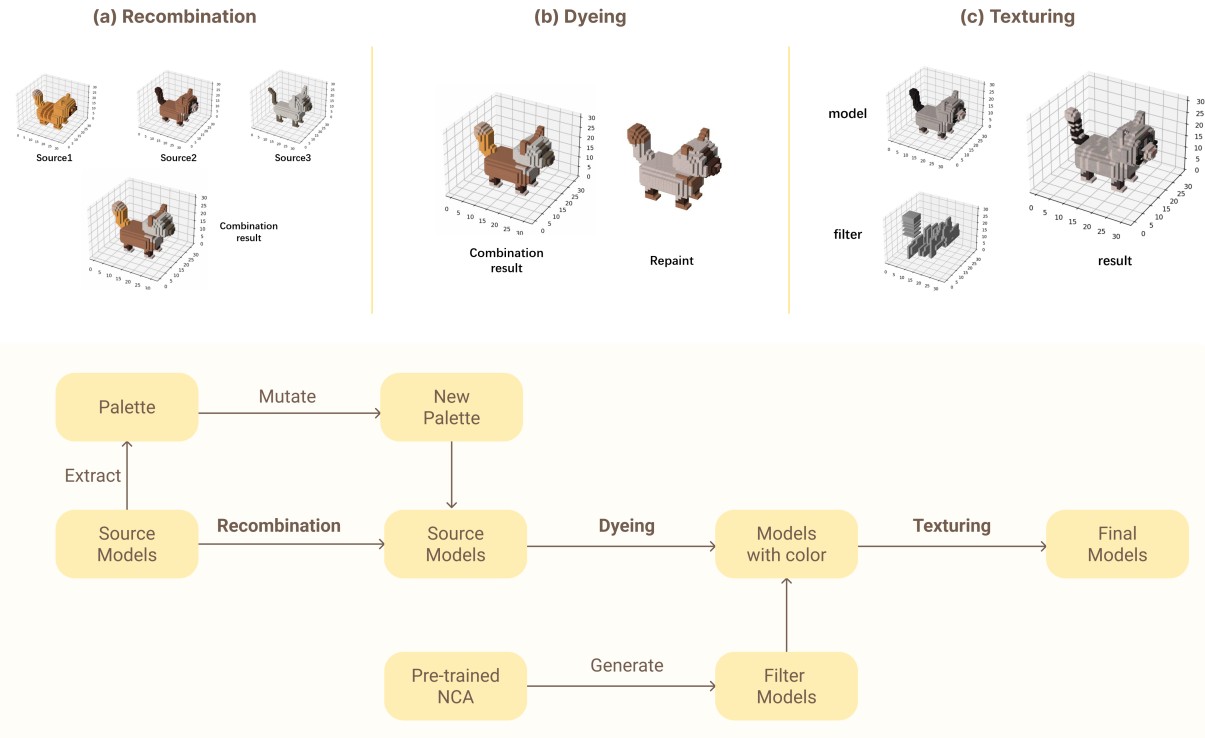

Figure 5: Generation Procedure

study aimed to explore how participants perceive virtual pets' personalities by observing their appearances using quantitative and qualitative methods.

## 6.1 Participants

We employed convenience sampling by posting our survey link through WeChat subscriptions. 57 participants (19 males, 33 females, 3 non-binaries, and 2 who preferred not to say) voluntarily completed the survey, different from the participants in study one. The age distribution of the participants consisted of participants aged 18-24 years (N=39) and 25-34 years (N=18). 47 participants had previous experience in owning real pets, while 41 had experience in owning virtual pets. Following the survey, 12 participants (6 males, 5 females, and 1 who preferred not to say) voluntarily participated in the interview session. The age distribution of the interviewees included participants aged 18-24 years (N=7) and 25-34 years (N=5). Of these interviewees, 9 had experience owning real pets, while 10 had experience owning virtual pets.

## 6.2 Measurement

The research employed an online survey consisting of four distinct parts, namely the IPIP-20 [17], the Motives for Online Gaming Questionnaire (MOGQ) [16], likability assessment of virtual pets, and demographic data collection. The IPIP-20 is a brief instrument for assessing FFM of personality traits using the International Personality Item Pool (IPIP) resources. This study utilized the questionnaire to identify participants' personalities, as our modified pet personality questionnaire in Study 1 shared four personality traits with the Big Five personality traits. The primary objective was to correlate participants' personality traits with their perceived personality of virtual pets to investigate the link between the two constructs. Both the IPIP-20 and MOGQ have demonstrated reliability and validity in measuring people's personalities and gaming behavior [16, 17].

Given the convenience sampling technique utilized to recruit participants, which may have reached a broad Chinese population, the study included validated Chinese translations of the IPIP-20 [39] and MOGQ [89]. In the likability assessment section, participants were presented with ten images, comprising three manually designed virtual pets (M1, M2, M3) and seven computer-generated ones (G1 to G7), refer to Fig. 6, and requested to rate their likability on a scale of 1 to 5. The demographic section of the survey collected data on participants' gender, age, education level, and pet-raising experience.

The follow-up interview consisted of questions parts and one card-sorting session in 45 minutes. The interview guide a series of open-ended questions about the virtual pet's personality, such as "What kind of personality do you think this virtual pet has?" and "What do you like and dislike about this virtual pet's appearance?" refer to the appendix for the interview outline. We created a personality mapping template for the card-sorting session, which consisted of four areas representing four personality traits and pictures of the ten virtual pets as cards that needed to be sorted in Figma.

## 6.3 Procedure

All participants completed a Qualtrics form that contained four test parts. After analyzing the survey data, we contacted twenty participants whose personalities and pet-keeping experiences are diverse via email. Twelve of them agreed to take part in the interview. We conducted the interviews through Voov Meeting. We informed participants of the study's purpose and procedures and obtained their written consent before the study began. Participants received 100 RMB as compensation.

## 6.4 Quantitative Results

The statistical results showed that appearance features played a significant role in participants' likability ratings of virtual pets,

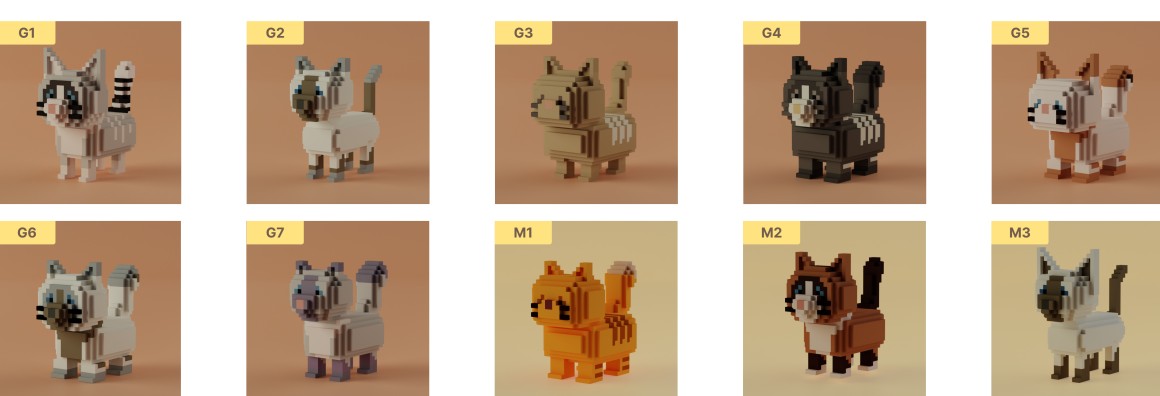

Figure 6: Generated Virtual Pets and Manual Designed Virtual Pets.

Table 1: Mean and SD of the Likability of Each Virtual Pet

| Virtual Pet ID | Mean | SD |
|---|---|---|
| M1 | 3.72 | 1.18 |
| M2 | 2.82 | 1.30 |
| M3 | 2.96 | 1.35 |
| G1 | 2.68 | 1.36 |
| G2 | 2.70 | 1.27 |
| G3 | 2.46 | 1.38 |
| G4 | 2.67 | 1.26 |
| G5 | 3.70 | 1.30 |
| G6 | 3.35 | 1.79 |
| G7 | 2.53 | 1.27 |

Table 2: Correlation Coefficient Between Participant's Features and Likability of Virtual Pets

| Scale | Feature | DV | Coef | p-value |
|---|---|---|---|---|
| IPIP-20 | Extraversion | G2 | -0.36 | 0.006* |
| MOGQ | Social | G1 | 0.41 | 0.002* |
| | Social | G6 | 0.26 | 0.05* |
| | Social | G7 | 0.29 | 0.03* |
| | Skill Development | G2 | 0.35 | 0.008* |
| | Skill Development | G7 | 0.37 | 0.005* |

specifically cobby cats with light-colored coats and decorations. Additionally, we identified significant correlations between certain personality and player-type traits and likability ratings of virtual pets.

### 6.4.1 Preference toward Cobby Cats with Light-colored Coat

We conducted a repeated-measures ANOVA to compare the likability ratings of ten virtual pets. Table 1 presents the descriptive statistics for these ratings, Fig. 6 illustrates each virtual pet. The ANOVA revealed a significant main effect for virtual pets ($F(9,81) = 12.02, p < .001$). To identify which virtual pets differed significantly from each other, we conducted post hoc tests using the Holm correction to adjust for multiple comparisons. The results showed that the likability ratings for G5 (a cobby cat with a wight coat and orange coloring on its ears, breast and tails) and M1 (a cobby cat with an orange coat with brown stripes on the back) were significantly higher than those for all other virtual pets except G6. In turn, the likability rating for G6 was considerably higher than those for G3 and G7, which received the lowest mean ratings. These findings suggest that appearance features significantly influence participants' likability ratings of virtual pets. Specifically, cobby cats with light-colored coats, such as G5, G6, and M1, were rated as more likable than other virtual pets. Additionally, color decorations on the fur, such as the orange coloring on the ears, tail, and breast of G5 and G6 and the brown stripes on the back of M1, also appeared to positively impact participants' likability ratings.

### 6.4.2 Effects of Personality and Player Type on Virtual Pet Likability Ratings

We examined the relationship between participants' personalities and player types and their ratings of virtual pets using Pearson

correlation. Before analysis, we checked the normality assumptions of variables using the Shapiro-Wilk test. The distributions of the 21 variables (10 likability, 5 personality traits, 6 player type traits) were assessed using the Shapiro-Wilk test, and the results indicated that more than half of the variables (N=16) were significantly non-normal ($p < 0.05$). Of the non-normal variables. Given the non-normality of all liabilities of virtual pets, non-parametric tests were used to assess the relationships between variables. The Spearman's rank correlation showed that five pairs of variables were significantly correlated: Extraversion and G2 ($p = 0.006$), Social and G1 ($p = 0.002$), Social and G6 ($p = 0.05$), Social and G7 ($p = 0.03$), Skill Development and G2 ($p = 0.008$), and Skill Development and G7 ($p = 0.005$). The correlation coefficient between Extraversion and G2 was negative ($Coef = -0.36$), while the correlation coefficients between the other five pairs were positive, ranging from 0.26 to 0.41 (see Table 2).

## 6.5 Qualitative Results of Interview

The qualitative results show a strong correlation between virtual pets' physical features and personalities attributed by users. Appearance plays a significant role in conveying their personalities, as participants identified specific personality traits based on design elements like body shape, skin color, facial features, and expressions. Additionally, users' personalities can influence their preferred pet personalities, explaining why some prefer pets with corresponding personalities. Most participants found the voxel-style pets more relaxing and easier to create with intricate details than the realistic-style pets. Overall, appearance significantly shapes users' perceptions of virtual pets, and their preferences are related to the emotions evoked by the pet styles.

### 6.5.1 Correlation between virtual pets' design elements and personalities attributed by users.

Through our study, we discovered that the appearance of our virtual pets was instrumental in conveying their personalities to participants.

Most participants (N=7) identified specific personality traits based on certain design elements, such as the shape of the pets' body and their skin color. For instance, participants associated warm, light-colored skin, fat and round body with agreeableness traits. One participant remarked, *"The pets' warm and light appearance made them feel tame and sweet, like dessert.* Seven participants (N=7) also noted that the facial features and expressions of the pets influenced their perceived personalities. For instance, one participant pointed out, *"the kitten's dark middle face gave it a neurotic look, which they associated with gloominess and neuroticism."* We have included a diagram in Fig.7 that showcases all the appearances mentioned in the interview and the traits related to the pets' personalities. In conclusion, our findings indicate that appearance significantly shapes users' perceptions of virtual pets.

### 6.5.2 Participants' Personalities were Related to Their Pet Choices

Our interview reveals a relationship between the personalities of participants' preferred pets and their personalities. Ten participants have similar personalities to the preferred pets (N=10). Eight of the participants admitted that they preferred pets with personalities similar to their own (N=8), while the rest made unconscious choices (N=2). According to the analysis, when participants liked their personalities, they preferred pets that were similar to their personalities (N=8). On the contrary, they do not like pets with similar personalities (N=2). For instance, P5 mentioned, *"Maybe it's because I'm impulsive, so I don't like (the pets that are impulsive)."* Thus, participants' personality preferences influence their choice of pets.

### 6.5.3 Comparison of Participants' Preferences for Voxel Style and Realistic Style

Most participants preferred the voxel style to the realistic style (N=8), while three preferred the realistic style (N=3), and two accepted both styles (N=2). Based on the analysis, participants' decisions on style preferences were related to the emotions evoked by the pet style. Participants who preferred the voxel style felt that it made them feel more relaxed (N=8), while the realistic style made them feel scared and overwhelmed (N=4). For instance, P11 explained, *"I think that this cat's eye and its overall appearance makes me feel Uncanny Valley."* Besides, although seven of the pets are machine-generated (G1 to G7), this does not affect the participants' preference for style. Participants could not distinguish between machine-generated pets and manually designed pets (N=4). Also, they believed that machine-generated pets basically had similar characteristics to real pets (N=4).

## 7 DISCUSSION

We conducted this study to investigate players' perceptions of virtual pets' personalities and discover the relationship between personalities and appearance. In the following sections, we discussed our findings through the results.

### 7.1 Perceiving Virtual Pet Personalities through Style and Representation

In our first study, we investigated how the style (voxel or realistic) and representation (static or animated) of virtual pets influence users' perceptions of their personalities. Our findings revealed that both factors significantly affected users' perception of virtual pet personalities. Specifically, participants perceived virtual pets with voxel style as friendlier, cuter, and more playful than those with realistic styles. This finding is noteworthy because it contrasts with previous research on similar personality ratings for realistic and cartoon avatars in virtual human characters [69]. We suggest that users' preference for the voxel style may be due to its association with an abstract and cartoonish aesthetic, which enhances the presentation

of pets' personalities. Our interview results further support this interpretation, as participants expressed a greater attachment to virtual pets with the voxel style, citing their agreeableness and cuteness, and the greater imagination space offered by the voxel style. In contrast, some participants found the realistic style to make virtual pets feel robotic and uncomfortable, reducing their emotional connection to them, a phenomenon known as the uncanny valley [56]. These findings underscore the significance of considering virtual pet style and representation in design, as they can impact users' perceptions of digital characters' personalities.

### 7.2 The Link between Virtual Pets' Personalities and Appearances

We aimed to explore the relationship between virtual pets' appearance and their perceived personality traits. Building upon previous research by Hanna Ekström [7], which suggests that visual cues such as shape and proportions can significantly influence how viewers perceive a character's personality traits, we designed six pet clusters with different visual cues to present various personality traits, as shown in Fig.1. Study 1 found that participants' perceptions of virtual pets' personalities aligned with our design intentions. Specifically, participants rated cat cluster 1 with a high agreeableness score and cat cluster 2 as more extroverted, consistent with previous research that suggests round and soft shapes are associated with friendliness and warmth. In contrast, angular and sharp shapes convey aggression and danger [7].

To further explore the relationship between virtual pets' appearance and perceived personality traits, we conducted study 2. Here, we utilized machine learning techniques to generate more voxel cat pictures and evaluated their personality presentation, as shown in Fig. 7. Our findings suggest that skin color is the most notable visual cue for describing voxel pets' personality traits. Participants perceived cats with warm-tone skin color as more friendly and sweet, while markings on a cat's face were associated with neuroticism. Additionally, participants linked a fat and round body shape with agreeableness and extroversion traits. In contrast, a towering body was associated with impulsiveness traits, and a small body was linked to neuroticism. Furthermore, we found that other parts of the virtual pets, such as the head, legs, and tails, could also provide visual cues for conveying personality traits.

Overall, our findings suggest that visual cues can significantly influence how virtual pets' personalities are perceived and that different parts of a virtual pet's appearance can provide valuable information for conveying personality traits. These results could be useful for designing virtual pets that accurately convey specific personality traits and enhance user engagement in future virtual pet game design.

### 7.3 Design Implications For Virtual Pet Characters Design

Our analysis of studies one and two leads to several design implications for virtual pet character design. These include considering players' preferences for virtual pets, designing with players' personalities, and developing interacting features in virtual pet games.

#### 7.3.1 Preference on Virtual Pets

The results of our study suggest that players generally prefer virtual pets in the form of cats and dogs, with some expressing interest in fantasy pets like dragons. Interestingly, our research also showed that participants preferred virtual pets with a neurotic personality and a cute appearance, with common traits including warm-tone skin colors, large eyes, and fat body shapes. These findings align with previous research that suggests people prefer dogs' features associated with the infant schema [35].

Additionally, our study found that the style and presentation of virtual pets had a significant impact on players' perceptions of their

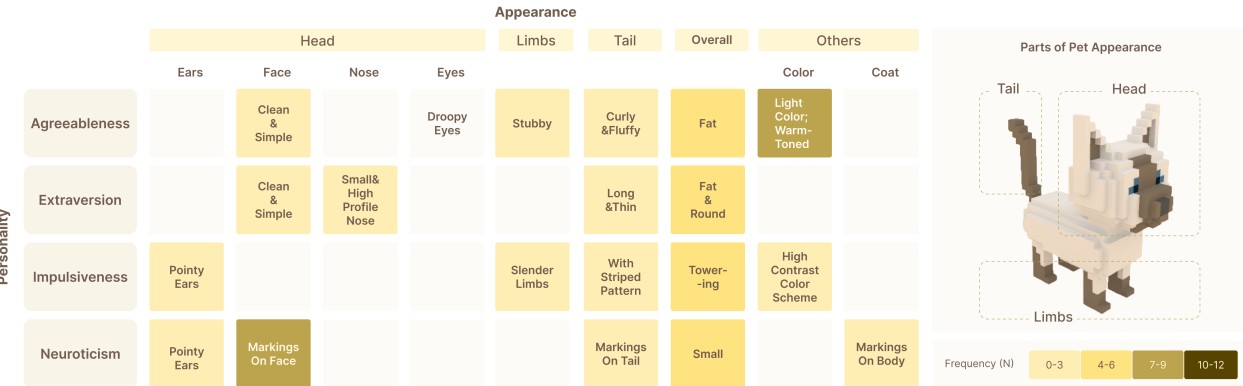

Figure 7: This figure depicts the relationship between the appearances of virtual pets and their perceived personalities. The left panel shows how different body parts of virtual pets relate to personality traits (A, E, I, N), with the frequency of each trait indicated by the corresponding color. The right panel displays a virtual pet image with labeled body parts for reference.

personalities. The majority of participants showed a keen preference for the voxel style, finding its abstract and cute appearance to be calming and potentially effective in reducing anxiety and depression compared to a realistic style. This result is consistent with previous research that suggests virtual animals' MR-based interaction can reduce mental stress and induce positive emotions [58]. However, our interview results revealed that the visual design of the virtual animal used in the previous study was realistic, which potentially cause players to experience the uncanny valley effect. In this phenomenon, a realistic but not quite natural appearance can cause unease or discomfort.

We also identified the most popular interaction schemes with virtual pets, such as talking, touching, and feeding, which can serve as a reference for future virtual pet game design. Additionally, our research showed that players prefer virtual pets to take on the role of companions rather than mentors or enemies, which differs from the suggestions made by previous researchers for non-player character roles in narrative settings.

These findings emphasize the importance of designers considering players' preferences for virtual pets' type, style, and interaction role when designing virtual pets. Designers should aim to create virtual pets with cute and endearing features while also incorporating traits that add depth to their personalities.

### 7.3.2 Incorporating Player's Personality and Virtual Pets' Personality in Designing Virtual Pets

While prior studies have examined pets' personalities, finding that owners were more satisfied with cats that were high in agreeableness and low in neuroticism [19], our research delved into the link between pet and owner personalities, focusing on virtual pets. Specifically, our study found that participants preferred virtual pets with personalities similar to their own, as indicated by the quantitative results of study two. Interestingly, our analysis also revealed that individual personality differences might influence how participants perceive and rate virtual pets. For instance, those with extraversion traits showed a preference for a virtual pet with slim legs and thin bodies (i.e., G2) that was more extroverted.

Our qualitative findings further supported this, showing that individuals with agreeableness traits in their personalities favored friendly and less aggressive virtual pets. Additionally, our results mirrored those of previous researchers, who noted a positive link between owner dominance and cat dominance, extraversion, and neuroticism [19]. However, unlike their work on natural pets, our study examined this relationship among virtual pets. Overall, our findings suggest that pet personality is an essential factor to consider in designing virtual pets and that personality traits of both the pet and owner may influence user preferences and satisfaction.

Our investigation into players' preferences was partly inspired by prior work identifying three user types based on preferences and gameplay styles in VR pet games [48]. In addition, we examined how individual differences in players' in-game behavior influenced their perception and ratings of virtual pets. We found that participants who played the game with a social or skill development purpose rated virtual pets with extraversion traits higher, as identified through interviews. These pets were characterized by cold-tone skin colors, small heads, and ears (G1). Our findings suggest that considering players' in-game models is a promising approach for designing virtual pet characters that align with users' preferences and engagement styles.

### 7.4 Controllable and Cost-Effective Model Generation using Recombination and Repainting to Improve User Response Data Quality

In the discussion section, we analyzed the effectiveness of our machine-generated pet pictures, which were used to collect user responses in our experiment. Our generator utilizes a recombination and repainting approach to produce high-quality results, and the method is relatively inexpensive compared to 3D generative neural networks. While generative neural networks are only minimally used for texturing due to insufficient training data, using them for random texturing can significantly reduce poor generation results. However, relying solely on traditional generation methods for shape and color can lead to less creative and predictable results. However, incorporating new colors and textures can make it difficult for people to identify the origin of the model's parts. In our study, we conducted a semi-structured interview in which participants were unaware that a machine generated the cat pictures. All participants noted the cat's body and other parts' excellent convergence.

### 7.5 Limitation and Future Work

This study has several limitations that we need to address in future work. Firstly, the limited training sample we used to generate cat pictures using machine learning techniques may have resulted in the lack of diversity in the appearance and personality traits of the voxel pets. To overcome this limitation, we plan to create and label diverse pets' body parts with different visual cues to better capture a broader range of personality traits.

Secondly, our design only focused on the static voxel style. It lacked animations and sound, which may have hindered the accurate perception of the pets' personalities in a more precise way. To improve the accuracy of personality perception and enhance the user experience, we plan to involve animated clips by rigging the voxel pet models and add sounds related to pet behaviors when designing virtual pets.

Furthermore, we plan to use our current voxel pet characters as artificial companions and integrate their personality traits to design an interactive virtual pet game. The game aims to reduce anxiety and stress levels as an intervention tool. It provides users with a fun and engaging way to interact with virtual pets, potentially improving their mental health and well-being.

In conclusion, although our study provides valuable insights into the link between virtual pets' appearances and their perceived personality traits, several limitations must be addressed in future work. By enriching our sample for generating voxel pets, involving animations and sound, and developing an interactive virtual pet game, we hope to provide users with a more authentic and engaging virtual pet experience.

## 8 CONCLUSION

In conclusion, our study aimed to address the gaps in current research on virtual pets and their potential for promoting mental health and enhancing skill development in individuals who cannot keep real pets. We focused on creating virtual pets with personality traits and exploring how players perceive their personalities. Our research found that appearance variations affect users' perceptions of virtual pet personalities. Players prefer virtual pets like cats and dogs with neurotic personalities and cute appearances. We also developed a novel method for a game character design that combined traditional methods with machine learning techniques. Our study provides several design implications for virtual pet character design. It highlights the potential of using voxel pets' appearances as virtual companions to enhance the mental well-being of young individuals by reducing anxiety levels through interactive engagement with virtual pets. Overall, our study contributes to the understanding of factors contributing to the development of personalities in different species and how we can design artificial companions that mimic and respond to these traits.

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

## A  SEMI-STRUCTURED INTERVIEW SCRIPT

### A.1  Semi-structured Interview of Study 1

Thank you for filling out our questionnaire earlier. We'd like to briefly talk with you about your past pet ownership experience and your thoughts on the personality and appearance of virtual pets. This interview is mainly divided into four parts. We will record the interview process, and some operations will be recorded on the screen. If you agree, let's continue.

First, let's recall how you filled out the questionnaire before. Here are the pictures and animations of pets you saw before: Pet Personality Comparison: (pictures for recall)

#### A.1.1  pet personality

1. We found you have to XX this attribute evaluation in XX, which means more XX. Do you think so?

2. What are the specific reasons that make you feel that way?

3. Regarding virtual pet cats/dogs, do you think animation is a better way to express the pet's personality than static images? And Where is it embodied?

4. Do you think virtual pet images (in the form of static and dynamic voxels) are better or worse at expressing a pet's personality than real pet photos?

5. You're looking at a picture of a cat/dog, but if you've had a cat/dog in your pet live, does that make a difference to your judgment?

   - If so, where are the main areas affected?
   - If not, why not?

#### A.1.2  Pet Type and Interaction

1. We found out that you used to have XXX, but the pet you want to have is still/is XXX. Can you tell us the reason?

2. The virtual pet you are looking forward to is XXX. Can you tell me why?

3. What characteristics do you think these virtual pets need? In appearance and personality?

4. You mentioned that if you can keep a virtual pet, the most important way to interact with him is XXX. Can you explain why you like this kind of interaction?

5. What do you find most appealing about this type of interaction?

6. Can you talk about the kind of interaction you want?

7. In addition, we also found that other ways of interacting are trendy. If you could choose the top three, what would you choose and rank them?

#### A.1.3  Pet Appearance

1. You describe your previous pet's personality as XXX. Do you think its physical characteristics are related?

   - If so, what specific characteristics (e.g., body shape, limbs, tail position, movement habits) indicate this personality trait?
   - If you don't feel connected, can you explain why?

2. Do you think there is a contrast between his appearance and his real character? Is that a big contrast?

3. We found that many people like pets that look cute. Do you agree?

   - If so, why do you like pets that look cute?
   - If not, why not?

4. What characteristics do you consider cute?

5. We've got some pictures and need your help choosing which ones you think are cute. That's great. We also want you to rank the cuteness of your picks.

#### A.1.4  Summary and Advice For Pet Design: Summary and advice for pet design:

1. Statistics show that the highest rating for our three pet pictures is XXX. Can you explain why?

2. Do you think our virtual pet characters can convey the pet's personality traits?

   - If so, can you tell me how you feel about it? (Color, movement, expression?) Which virtual pet do you prefer to your previous pets? Why?
   - If not, could you tell us the specific reason?

3. What do you think of our virtual pet characters?

4. Any suggestions on how to improve the design of pet characters or animations?

**A.2 Semi-structured Interview of Study 2**

Thank you for participating in this interview. We would like to chat briefly with you about your thoughts on the pet images we designed, as well as a deeper understanding of some of your suggestions for anxiety-relieving applications. The interview will be divided into five parts, about 60 to 90 minutes. We will tape and video the interview. If you agree, let's continue.

1. We found that your pet evaluation is: XXX is the highest score. Can you explain why? classifying pets' personalities, we found that the pet you rated highest was XX personality. Can you explain why?

2. We have prepared some pictures of voxel pet. Please put them in the best area according to the images you see. We found xxx, could you please explain why?

3. What do you think is the difference between xx?

## A.2.1 Design

1. Compared with the painting style of real and voxel pet, which do you prefer? Can you expand on that?

2. Which image do you prefer compared to the real and voxel pet? Can you expand on that?

3. What is your favorite part of the Voxel pet's look?

   - What's your least favorite part?
   - Can you expand on that?

4. Compared with the real and voxel pet, which image do you find more relaxing and relaxing?

5. Which do you prefer compared to the real pet and voxel pet scenes? Can you expand on that?