# OpenReview forum: "“I Keep Sweet Cats In Real Life, But What I Need In The Virtual World Is A Neurotic Dragon": Virtual Pet Designs With Personality Patterns"
_graphicsinterface.org/Graphics_Interface/2023/Conference_SD — Submitted to GI 2023 - second deadline_

### Official Review · Reviewer_X3jf · 2023-04-23
**Unfortunately, this submission has not been anonymized (as was specified in the call for papers/submission guidelines), however, I have provided a review nonetheless for the author's benefits.**

**Rating:** 3
**Confidence:** 3

**Review:**

In this paper, the authors conducted a study to investigate people's perceptions of virtual pets' personalities and appearances through two user studies. Their findings suggest that some styled pets better preferenced agreeableness than realistic representations, and that users preferred virtual pets with whom personalities were shared. The authors results provide insights for game designers and researchers to build upon in the design of future pet--based games and understand how people perceive virtual pets based on their appearance and behaviour. Their findings include: 1) style and presentation significantly affect users' perceptions of virtual pet personalities, and 2) visual cues can significantly influence how virtual pets' personalities are perceived. Through their work, the authors contribute: 1) an evaluation of their designed virtual pet characters with pre-defined personality traits from a user study, 2) the combination of traditional and machine learning methods for generating virtual pets' appearances, and 3) insights towards the potential for using voxel pets' appearances as virtual companions to increase the mental well-being of young individuals through interaction and engagement.

Positives/Pros:
- The first paragraph of the introduction provides a strong and broad introduction to motivate the further exploration of virtual pets.
- Results are clearly articulated and demonstrated through the data. However, while 4.4 (Results) does a good job of articulating specific results with statistics, for the average reader, explanation of these individual factors and the outcomes would be beneficial.
- Within sections, there are attempts at tying the current discussions back to earlier results or findings. However, this tends to also come off as repetition at times and can likely be lessened in order to also shorten the length of the paper overall.
- I really appreciate and enjoyed examining Figure 7! This figure does a great job of visually displaying the results as they relate to the different metrics for personality and appearance. I wonder if the authors could reframe their results and discussions section based on this figure and highlight the interesting patterns that emerged?

Questions/Concerns:
- The paper is not anonymized as per the paper submission guidelines/call for papers for GI 2023. In future submissions, please be extra cautious of all submission policies and guidelines to ensure maximum benefit. Further, in section 6.2, the authors note "…which may have reached a broad Chinese population, the study included validated Chinese translations…". This mention of a particular demographic does also risk revealing or reducing the anonymity of the authors, institution, origin, or location from which the paper and study originate.
- Some sections of the paper read to be less polished than others. I would encourage the authors to proofread the entire paper again to catch minor grammatical or punctuation errors, and remove any repetition that might be present within sentences or amongst sections. There are also points (for example at the beginning of 6.5.3) where the counts do not add up to the participant number, causing confusion for readers.
- There's some confusions when it comes to participants - for example, it states that 24 participants had the experience of keeping real pets, but only 14 are accounted for in the provided count (i.e., 7 had dogs and 7 had cats). What about the other 10?
- In 4.3 (Process), there are missing details about how the analysis was carried out, how participants were determined, who completed this procedure from the research team, and how answered were deemed "consistent" or "contrary" - what was the baseline?

Overall, throughout this paper there are a series o inconsistencies in formatting (for spacing, grammar, punctuation, participant quoting), and sections that would need to undergo further proofreading in order to make this work publication ready. Additionally, the varied and inconsistent lengths of sentences and differing levels of formality in language make this paper overall a bit tougher to parse and follow.

---

### Official Review · Reviewer_Z3qr · 2023-05-01
**Some presentation issues that are largely overcome by a well motivated, well executed piece of research**

**Rating:** 7
**Confidence:** 3

**Review:**

The authors of this paper investigate the design of virtual pets with personality patterns through a design phase then a study of those designed virtual pets.
I do not know much about virtual pets, but I did enjoy reading this paper quite a bit. I learned quite a few things about this sub-area, found good motivation for the work, and learned about interesting findings regarding how people perceive the personality of virtual pets that I think is of value to the HCI/social computing research community. The design process is well described and rooted in the literature; the pet generation is grounded in the results of a formative study that consists of a survey and an interview. The evaluative study is well designed and well executed. The images are well crafted and informative. I cannot comment on the novelty of the approach given my little knowledge of the topic. Figure 7 summarizes the findings is a very neat and useful way. Despite the sample size for the studies being relatively small, the results have value and open the door to more work in the area, as acknowledge the authors.

For all these reasons, I am in favor of accepting the paper at GI (with the limit of the non-anonymity problem).

On the down side, the paper would benefit from several improvements, and I hope the authors consider these when finalizing their paper (or resubmitting it someplace else):

0. The paper was not anonymized; however I still reviewed the paper as is, given that I am not in conflict with any of the authors/institutions listed.
1. I would have liked to read more details about the qualitative data analysis for Study 1. No methodological details are provided.
2. There are many typos/mlanguage mistages in the paper that must be addressed (e.g., missing labels, wrong tense like in "In the following sections, we discussed our findings", etc..). Overall, the paper writing presentation is sloppy. The authors would need to complete a thorough editing of the write-up if the paper was to be accepted.

---

### Official Review · Reviewer_VB4K · 2023-05-01
**Virtual Pets appearance and type of media affects perception of their personality traits.**

**Rating:** 3
**Confidence:** 3

**Review:**

Positives

+ Virtual pets make sense, and I do believe that their use will be increasing in the short term
+ The paper contains the description of two studies and a system to generate 3D pets (minecraft styles), based on neural networks

The paper also has significant problems, that make me recommend rejection at this stage. Specifically:

1. The purpose of the studies and the paper is not well articulated. Although the authors do convey that there is a gap in the knowledge of how the visual appearance of different representations of pets would affect the perception of their personality, it is not really clear how having that knowledge could be really be helpful. I am skeptical that knowing, for example, that representations of animals/pets that have lighter fur, will help in any way design games or systems to keep VR pets that are more effectively at their objectives (from the paper, e.g., reducing stress).
2. The text in the study results and discussion are overgeneralizations of the results in the study. This affects many aspects of the claims made about the two studies. There are many examples of this. One of the problems is that it is difficult to generalize from just a few pets; even if the participants made some comments, it is doubtful that most people will react in the same way. Moreover, many of the results are likely the results from characteristics of the pets that have not been controlled (i.e., they are likely confounds).
3. The description of the experiments is not sufficiently detailed to assess the quality of the study. This is compounded by the lack of clarity in the writing. For example, it is often not clear whether the "real" alternative of the animals were photos or otherwise generated versions of the pets.

Overall this might be good work, but a lack of clear motivation, the overgeneralization of claims, and the lack of clarity in the design of the studies makes me suggest that this submission is not above the bar for acceptance in its current form.